# AST-T5: Structure-Aware Pretraining for Code Generation and Understanding

## ABSTRACT

Large language models (LLMs) have made significant advancements in code-related tasks, yet many LLMs treat code as simple sequences, neglecting its structured nature. We introduce AST-T5, a novel pretraining paradigm that leverages the Abstract Syntax Tree (AST) for enhanced code generation, transpilation, and understanding. Using dynamic programming, our AST-Aware Segmentation retains code structure, while our AST-Aware Span Corruption objective equips the model to reconstruct various code structures. Unlike other models, AST-T5 avoids intricate program analyses or architectural changes, so it integrates seamlessly with any encoder-decoder Transformer. Evaluations show that AST-T5 consistently outperforms similar-sized LMs across various code-related tasks. Structure-awareness makes AST-T5 particularly powerful in code-to-code tasks, surpassing CodeT5 by 2 points in exact match score for the Bugs2Fix task and by 3 points in exact match score for Java-C# Transpilation in CodeXGLUE. Our code and model are publicly available at *https://anonymized*.

## 1 INTRODUCTION

We have witnessed the transformative impact of large language models (LLMs) on various aspects of artificial intelligence in recent years (Brown et al., 2020; Ouyang et al., 2022; Touvron et al., 2023), especially in code generation and understanding (Feng et al., 2020; Wang et al., 2021; Rozière et al., 2023). By pretraining on massive code corpora such as the GitHub corpus, LLMs learns rich representations, thereby becoming powerful tools for various downstream applications, including text-to-code generation (Chen et al., 2021a; Austin et al., 2021; Iyer et al., 2018), code-to-code transpilation (Lu et al., 2021; Lachaux et al., 2020; Tufano et al., 2019), and code understanding (mapping code to classification labels) (Zhou et al., 2019; Svajlenko et al., 2014).

Despite these impressive advances, most existing models interpret code as mere sequences of sub-word tokens, overlooking its intrinsic structured nature. Prior research has shown that leveraging the Abstract Syntax Tree (AST) of code can significantly improve performance on code-related tasks (Guo et al., 2021; Tipirneni et al., 2023). Some studies also use code obfuscation during pretraining to teach models about abstract code structures (Roziere et al., 2021; Wang et al., 2021). However, these models often rely on computationally expensive processes like Control-Flow Analysis (CFA), obfuscation, or even actual code execution. Such dependency limits their scalability and imposes stringent conditions like code executability. Consequently, these methods may struggle with real-world code, especially in intricate languages like C/C++, where comprehensive analysis remains elusive.

In this study, we propose AST-T5, a pretraining paradigm that leverages the Abstract Syntax Tree (AST) structure of code. The key contribution in AST-T5 is a simple yet effective way to exploit code semantics, without the need to run expensive program analysis or execution. Using a lightweight, multi-language parser called Tree-sitter[1], our approach has broad applicability across all syntactically well-defined programming languages. After we parse code into ASTs, we use a dynamic programming-based segmentation algorithm for AST-aware code segmentation to maintain the structural integrity of the input code. Using our novel AST-Aware Span Corruption technique, the model is pretrained to reconstruct various code structures, ranging from individual tokens to entire function bodies. Together, our approach offers three key advantages: (1) enriched bidirectional

---

[1] https://tree-sitter.github.io/tree-sitter/

Figure 1: Comparison of AST-Aware Subtree Corruption and Vanilla T5 using a Python factorial function. Both methods replace masked spans with sentinel tokens (special tokens added to the vocabulary, shown as [X], [Y], and [Z] in the figure), with output sequences containing the original masked tokens. Inputs and targets are shown in byte-pair encoding (BPE); for instance, "factorial" is encoded into "fact" and "##orial". Unlike Vanilla T5, which masks random spans without considering code structure, our approach specifically targets spans aligned with AST subtrees, like expressions and statements.

encoding for improved code understanding, (2) the ability to coherently generate code structures, and (3) a unified, structure-aware pretraining framework that boosts performance across a variety of code-related tasks, particularly in code transpilation.

In addition, other than our specialized AST-aware masking approach, AST-T5 introduces no architecture changes or additional heads, and our pretraining objective remains the same as Vanilla T5. This compatibility enables seamless integration of our model as a drop-in replacement for any T5 variant.

In our experiments, AST-T5 consistently outperforms baselines in code generation, transpilation, and understanding tasks. Through controlled experiments, we empirically demonstrate that these advancements are attributed to our AST-aware pretraining techniques. Notably, AST-T5 not only outperforms similar-sized models like CodeT5 and CodeT5+ across various benchmarks but also remains competitive with, or occasionally even exceeds, the performance of much larger models using the HumanEval dataset. Furthermore, the inherent AST-awareness of AST-T5 offers unique advantages in structure-sensitive tasks, such as code-to-code transpilation and Clone Detection, highlighting its effectiveness at capturing the structural nuances of code.

## 2 RELATED WORK

**Language Models for Code.** Language models (LMs) extended their use from NLP to code understanding and generation. Encoder-only models generally excel in code understanding when fine-tuned with classifiers (Feng et al., 2020), while decoder-only models are optimized for code generation through their autoregressive nature (Chen et al., 2021a; Fried et al., 2023; Nijkamp et al., 2023). However, these models can falter outside their primary domains of expertise or require increased resources for comparable outcomes. Our work focuses on encoder-decoder models, aiming to efficiently balance performance in both understanding and generation tasks without excessive computational demands.

**Efforts Toward Unified Models.** Extending NLP models like BART (Lewis et al., 2019) and T5 (Raffel et al., 2020), several studies have developed encoder-decoder architectures, such as

PLBART (Ahmad et al., 2021) and CodeT5 (Wang et al., 2021), to perform well in diverse code-related tasks. Although these models show broader utility, they struggle with generating coherent, executable code in complex scenarios like HumanEval (Chen et al., 2021a). CodeT5+ (Wang et al., 2023) seeks to address this limitation through an intricate multi-task pretraining strategy across five objectives. In contrast, our proposed model, AST-T5, uses a novel AST-Aware pretraining paradigm to become a unified model capable of generating fluent code and maintaining superior performance in code understanding tasks. Moreover, AST-T5 is more streamlined, because it only uses a single pretraining objective.

**Leveraging Code Structure in Pretraining.** Code differs from natural language in two key aspects: its executability and strict structural syntax. Previous research leveraged execution traces for improving model performance (Chen et al., 2018; 2021b; Shojaee et al., 2023), but this approach faces scalability challenges when applied to large, web-crawled code datasets used in pretraining. Regarding code's structured nature, various studies have integrated syntactic elements into neural network models. Li et al. (2018), Kim et al. (2021) and Zügner et al. (2021) add AST-Aware attention mechanisms in their models, while Alon et al. (2020) and Rabinovich et al. (2017) focus on modeling AST node expansion operations rather than traditional code tokens. In parallel, Guo et al. (2021) and Allamanis et al. (2017) explore DFG-Aware attention mechanisms and Graph Neural Networks (GNNs), to interpret code based on its Data Flow Graph (DFG). StructCoder (Tipirneni et al., 2023) enriches the code input by appending AST and DFG as additional features. These methods, however, necessitate parsing or static analysis for downstream tasks, which is less feasible for incomplete or incorrect code scenarios like bug fixing.

Our work, AST-T5, aligns with methods that utilize code structure only in pretraining, like DOBF (Roziere et al., 2021) and CodeT5 (Wang et al., 2021), which obfuscate inputs to force the model to grasp abstract structures. Our approach uniquely diverges by using AST-driven segmentation and masking in T5 span corruption during pretraining. This novel approach offers a more refined pretraining signal compared to structure-agnostic T5, equipping our model to proficiently encode and generate semantically coherent code structures.

## 3 METHOD

In this section, we present AST-T5, a novel pretraining framework for code-based language models that harnesses the power of Abstract Syntax Trees (ASTs). First, AST-T5 parses code into ASTs to enable a deeper understanding of code structure. Leveraging this structure, we introduce AST-Aware Segmentation, an algorithm designed to address Transformer token limits while retaining the semantic coherence of the code. Second, we introduce AST-Aware Span Corruption, a masking technique that pretrains AST-T5 to reconstruct code structures ranging from individual tokens to entire function bodies, enhancing both its flexibility and structure-awareness.

### 3.1 PARSING CODE INTO ASTS

Unlike traditional language models on code that handle code as simple sequences of subword tokens, AST-T5 leverages the Abstract Syntax Tree (AST) of code to gain semantic insights. For parsing purposes, we assume the provided code is syntactically valid—a reasonable assumption for tasks like code transpilation and understanding. Instead of the often computationally-intensive or infeasible methods of Control-Flow Analysis (CFA) or code execution (Guo et al., 2021; Tipirneni et al., 2023), our method only demands the code to be parsable. We use Tree-sitter, a multi-language parser, to construct the ASTs, where each subtree represents a consecutive span of subword tokens, and every leaf node represents an individual token.

### 3.2 OUR AST-AWARE SEGMENTATION

In this subsection, we describe our AST-Aware Segmentation method, which splits lengthy code files into chunks in a structure-perserving manner.

**Segmentation in language model pretraining** is a critical yet often overlooked aspect. Transformer LMs impose token limits on input sequences, making segmentation essential for fitting these

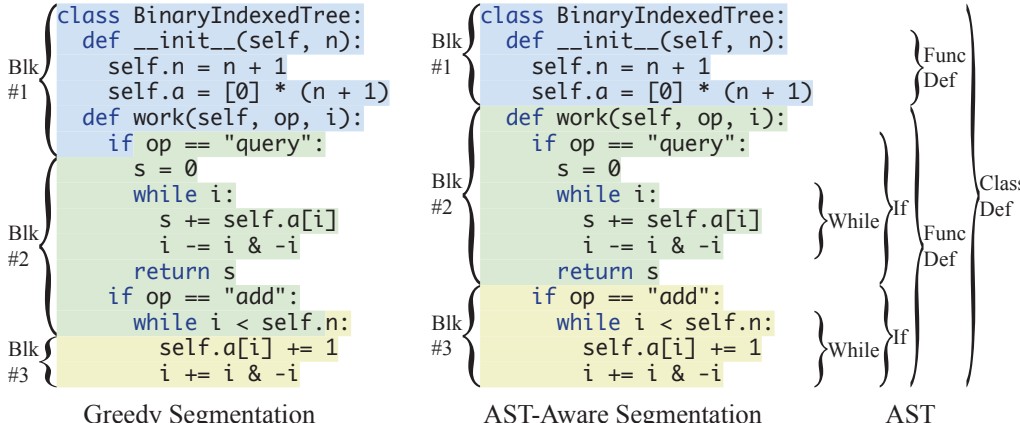

Figure 2: Comparison between Greedy Segmentation and AST-Aware Segmentation: For a 112-token code example with max_len set at 48, Greedy Segmentation places the first 48 tokens in Block 1, the next 48 tokens in Block 2, and the remaining in Block 3, disrupting the structural integrity of the code. In contrast, AST-Aware Segmentation uses a dynamic programming algorithm to smartly partition the code, aligning with boundaries of member functions or major function branches, thereby preserving the code's structure. The accompanying AST, with some levels pruned for clarity, corroborates that these segmentations indeed coincide with key subtree demarcations.

inputs within the max_len constraint. A naive approach is Greedy Segmentation, where each chunk, except the last, contains exactly max_len tokens Figure 2 (Left). This strategy has been widely adopted in previous works, such as CodeT5 (Wang et al., 2021).

Research in NLP by Liu et al. (2019) underscores that segmentation respecting sentence and document boundaries outperforms the greedy strategy. Given programming language's inherently structured nature, which is arguably more complex than natural language, a more sophisticated segmentation approach is even more important. However, this area remains largely unexplored.

**AST-Aware Segmentation** is our novel approach designed to preserve the AST structure of code during segmentation. Unlike Greedy Segmentation, which can indiscriminately fragment AST structures, our method strategically minimizes such disruptions. As illustrated in the example in Figure 2, Greedy Segmentation leads to nine instances of AST breaks—between Block 1 and Block 2, it breaks If, FuncDef, and ClassDef; between Block 2 and Block 3, it breaks Attr, BinaryExpr, While, If, FuncDef, and ClassDef. In contrast, our AST-Aware approach results in only three breaks: between Block 1 and Block 2, it breaks ClassDef, and between Block 2 and Block 3, it breaks FuncDef and ClassDef.

To identify optimal partition boundaries, we use a dynamic programming (DP) algorithm:

1. We construct an array cost, where cost[i] denotes the number of AST-structure breaks that would occur if partitioning happened right after token $i$. This array is populated by traversing the AST and incrementing cost[l..r - 1] by 1 for each span $[l, r]$ associated with an AST subtree.

2. We define a 2-D array dp, where dp[k, i] represents the the minimum total number of AST-structure breaks when $k$ partitions are made for the first $i$ tokens, ending the last partition right after the $i$-th token. The state transition equation is:

$$\text{dp}[k, i] = \text{cost}[i] + \min_{i-\texttt{max\_len} \leq j < i} \text{dp}[k-1, j] \tag{1}$$

3. While the naive DP algorithm has a quadratic time complexity $O(n^2)$ relative to the code file length $n$, it can be optimized to $O(n^2/\texttt{max\_len})$ by employing a monotonic queue for sliding-window minimum calculations. This allows for efficient computation across most code files. The pseudocode of the optimized dynamic programming algorithm is shown in Algorithm 1. See Appendix A.2 for details about complexity calculations.

4. The algorithm outputs the partition associated with dp[k_min, n], where k_min = $\arg\min_k(dp[k, n])$, as the most optimal partition.

---

**Algorithm 1** Dynamic Programming in AST-Aware Segmentation

```
1   # n: the length of the code file (number of tokens)
2   # m: the max number of segments; approximately n / max_len
3   for k in range(1, m + 1):
4       q = Queue()  # double ended queue
5       for i in range(1, n + 1):
6           while q.nonempty() and q.left() < i - max_len:
7               q.pop_left()  # pop indices before i - max_len
8           while q.nonempty() and dp[k - 1, q.right()] > dp[k - 1, i - 1]:
9               q.pop_right()  # maintain monotonicity of values
10          q.push_right(i - 1)  # Push i - 1
11          best_j = q.left()  # guaranteed to have the smallest value
12          prev[k, i] = best_j
13          dp[k, i] = cost[i] + dp[k - 1, best_j]
```

---

In comparing AST-Aware Segmentation with Greedy Segmentation—using the example in Figure 2—we find that the former presents more coherent code segments to the model during pretraining. Conversely, the latter introduces noisy partial expressions near partition boundaries. Consequently, AST-Aware Segmentation not only optimizes the pretraining process but also reduces the mismatch between pretraining and downstream tasks, which often involve complete function definitions as inputs.

### 3.3 Pretraining with Span Corruption

The pretraining task of AST-T5 is based on *span corruption*, a well-established method for pretraining Transformer encoder-decoder models (Raffel et al., 2020). In this approach, 15% of the input tokens are randomly masked and replaced by unique "sentinel" tokens, distinct within each example. Each unique sentinel token is associated with a specific ID and added to the model's vocabulary.

During pretraining, the encoder processes the corrupted input sequence. The decoder's objective is to reconstruct the dropped-out tokens based on the encoder's output representations. Specifically, the target sequence consists of the masked spans of tokens, demarcated by their corresponding sentinel tokens. This framework effectively trains the model to recover the original text from a corrupted input. Figure 1 (Left) illustrates an example of the input-output pair for span corruption.

### 3.4 Our AST-Aware Subtree Corruption

---

**Algorithm 2** Subtree Selection in AST-Aware Subtree Corruption

```
1   def mask_subtree(t: ASTNode, m: int): # mask m tokens in subtree t
2       ordered_children = []
3       m_remaining = m  # distribute m tokens among children of t
4       for child in t.children:
5           if child.size > theta:  # a hyperparameter to control granularity
6               m_child = m * (child.size / t.size)  # same mask ratio
7               mask_subtree(child, m_child)  # mask recursively
8               m_remaining -= m_child
9           else:
10              ordered_children.append(child)
11      weighted_shuffle(ordered_children)
12      for child in ordered_children:  # greedy allocation
13          m_child = min(m_remaining, child.size)
14          mask_subtree(child, m_child)
15          m_remaining -= m_child
```

---

AST-T5 augments the traditional span corruption paradigm by incorporating AST-awareness. Rather than arbitrarily masking consecutive token spans, AST-T5 masks code spans corresponding to AST subtrees, ranging from individual expressions to entire function bodies.

**Subtree Masking.** We use a recursive algorithm, outlined in Algorithm 2, to traverse the AST and select subtrees for masking. The algorithm aims to fulfill two goals:

1. Introduce sufficient randomness across training epochs to enhance generalization.
2. Control the masking granularity via a tunable hyperparameter $\theta$ (named `theta` in Algorithm 2, Line 5).

The "mask quota" $m$ denotes the number of tokens to be masked in a subtree rooted at node $t$. The size of a subtree corresponds to the number of tokens it encompasses, derived from the cumulative sizes of its children. For larger subtrees that exceed the size threshold $\theta$, masking is applied recursively (Lines 5-8). Meanwhile, smaller subtrees undergo a weighted shuffle, and the quota $m$ is then apportioned among $t$'s children in a greedy fashion according to the shuffled order (Lines 11-15). The weights for shuffling are determined by a heuristic function on the size of each child, such that masking probabilities are distributed uniformly across leaf nodes. To create a subtree mask for an AST rooted at $t$ with a mask ratio $r$ (e.g., 15% or 25%), one can use $\mathrm{mask\_subtree}(t, \lfloor |t| \cdot r \rfloor)$.

The parameter $\theta$ controls the granularity of masking. For example, with $\theta = 5$, the algorithm has a high probability to mask individual tokens and short expressions. As $\theta$ increases to 20, the algorithm is more likely to mask larger constructs such as statements. When $\theta = 100$, the probability increases for masking structures like loops or entire function bodies. To foster diverse training scenarios, $\theta$ is randomly sampled within a predefined range (e.g., 5 to 100) for each training example. This allows the pretraining framework to inherently accommodate tasks as varied as single-token completion to full function body generation from a given signature.

The subtree masking strategy is the primary distinction between our AST-Aware Subtree Corruption and the Vanilla T5 Span Corruption, as illustrated in Figure 1. While conventional T5 variants mask random token spans, with an average span length of 3 (Raffel et al., 2020) and neglecting code structures, our method targets the masking of AST subtrees, potentially encompassing up to 100 tokens. This equips AST-T5 for generation of various code structures coherently.

**Pretraining Objective.** Except for the strategy used to select masked tokens and the segmentation strategy described in Section 3.2 , our approach adheres to the workflow described in Section 3.3. Once subtrees are selected for masking and replaced with sentinel tokens, the encoder processes this modified input. Subsequently, the decoder is tasked with reconstructing the original tokens within the masked subtrees. A side-by-side comparison between our approach and the Vanilla Span Corruption in T5 is presented in Figure 1.

## 4 EXPERIMENTAL SETUP

**Model Architecture.** AST-T5 has an architecture similar to T5$_{\mathrm{BASE}}$ (Raffel et al., 2020), comprising a 12-layer encoder and a 12-layer decoder, where each layer has 768 dimensions and 12 attention heads. In total, the model has 226M parameters.

**Pretraining.** AST-T5 is pretrained on a mixed corpus consisting of code and natural language. Code is sourced from "GitHub repositories" dataset on Google BigQuery, which includes all code files from repositories with open-source licenses permitting redistribution. For NL, we use Wikipedia and OpenWebText, following Liu et al. (2019). Our corpus consists of 408 GB of code and 64 GB of text, smaller than the corpus used by CodeT5 (Wang et al., 2021) and CodeT5+ (Wang et al., 2023). Detailed statistics are provided in Appendix A.3.

Each code file is first parsed into its AST using the Tree-Sitter multi-language parser, and then tokenized with byte-level Byte-Pair Encoding (BPE) using a 64k BPE token vocabulary. Following AST-Aware Segmentation, these files are partitioned into chunks of 1,024 tokens. Our model is pretrained using the AST-Aware Subtree Corruption objective for 524 billion tokens (1,024 tokens per sequence, 1,024 sequences per batch, and 500k steps). For each training example, we apply AST-Aware Subtree Corruption of it is code, or apply Vanilla T5 Span Corruption of it is natural language. For code, the threshold, $\theta$, is uniformly sampled from 5 to 100. Pretraining uses PyTorch, Fairseq[2] and FlashAttention (Dao et al., 2022) and is conducted on 8 nodes, each with 8x NVIDIA A100 40GB GPUs. Further pretraining hyperparameters are detailed in Appendix A.4.

---

[2]https://github.com/facebookresearch/fairseq

Table 1: Performance comparison of various pretraining configurations for downstream tasks. Each row represents a sequential modification applied to the model in the previous row. Metrics include "Pass@1" rate for HumanEval, "Exact Match" rate for CONCODE, Bugs2Fix (for "Small" and "Medium" code lengths splits), and Java-C# transpilation (both Java-to-C# and C#-to-Java). F1 score is used for Clone Detection, and Accuracy for Defect Detection, consistent with prior studies.

| | Generation | | Transpilation | | Understanding | | |
|---|---|---|---|---|---|---|---|
| **Pretraining Config** | **HumanEval** | **Concode** | **Bugs2Fix** | **Java-C#** | **Clone** | **Defect** | **Avg** |
| T5 | 5.2 | 18.3 | 21.2/13.8 | 65.5/68.4 | 96.9 | 64.1 | 44.2 |
| + AST. Segmentation | 7.2 | 20.2 | 22.5/15.1 | 66.3/69.3 | 98.3 | 65.9 | 45.7 |
| + AST. Subtree Corrupt | 9.6 | 22.1 | 23.3/**16.5** | 67.3/72.2 | **98.6** | **66.0** | 47.0 |
| + Mask 25% (AST-T5) | 12.8 | **22.9** | **23.8**/16.1 | **68.9/72.3** | **98.6** | 65.8 | **47.7** |
| + Mask 50% | **13.0** | 22.0 | 21.9/15.0 | 66.5/70.1 | 97.1 | 64.2 | 46.2 |

**Evaluation.** We evaluate AST-T5 across three types of tasks: text-to-code generation, code-to-code transpilation, and code understanding (classification). Our evaluation encompasses tasks from the CodeXGLUE meta-benchmark (Lu et al., 2021) and also includes HumanEval (Chen et al., 2021a) and MBPP (Austin et al., 2021). Details about the benchmarks are shown in Appendix A.5.

We finetune AST-T5 on the training datasets of all downstream tasks, adhering to the methodology by Raffel et al. (2020). For the HumanEval task, which lacks its own training dataset, we use CodeSearchNet (Husain et al., 2020), aligning with the approach of Wang et al. (2023). The prompt templates for finetuning are constructed using the PromptSource framework (Bach et al., 2022). The finetuning takes 50k steps, with the peak learning rate set at 10% of the pretraining learning rate. All other hyperparameters from pretraining are retained without further adjustments, and we train only one finetuned model. During inference, rank classification is employed for code understanding tasks and beam search for generative tasks, following Sanh et al. (2021). We evaluate our model on the test set using five prompt templates for each task and report the average performance.

**Baselines.** We first benchmark AST-T5 against our own T5 baselines to ensure a controlled comparison. All models share identical Transformer architectures, pretraining data, and computational settings, differing only in the use of AST-Aware Segmentation and Subtree Corruption techniques by AST-T5. This setup directly evaluates the efficacy of our proposed methods.

We further benchmark AST-T5 against other language models for code-related tasks. These include decoder-only models such as the GPT variants (Brown et al., 2020; Chen et al., 2021a; Wang & Komatsuzaki, 2021), PaLM (Chowdhery et al., 2022), InCoder (Fried et al., 2023), and LLaMa (Touvron et al., 2023). We also compare with encoder-decoder models, including PLBART (Ahmad et al., 2021), CodeT5 (Wang et al., 2021), StructCoder (Tipirneni et al., 2023), and CodeT5+ (Wang et al., 2023). Notably, CodeT5$_{\text{BASE}}$ and CodeT5+ (220M) closely resemble our model in terms of architecture and size, but AST-T5 distinguishes itself with its AST-Aware pretraining techniques.

## 5 EVALUATION RESULTS

In this section, we evaluate AST-T5 across multiple benchmarks. First, we analyze the contributions of each component within our AST-aware pretraining framework through controlled experiments. Next, we benchmark AST-T5 against existing models in prior work.

### 5.1 PRETRAINING PROCEDURE ANALYSIS

In this subsection, we analyze the key components that contribute to the pretraining of AST-T5 models. Holding the model architecture, pretraining datasets, and computational environment constant, we sequentially add one component at a time to a T5 baseline trained on code, culminating in our finalized AST-T5 model. Table 1 presents the experimental results. These results show that:

**AST-Aware Segmentation enhances code language models.** A comparison between the first two rows of Table 1 shows that the model trained with AST-Aware Segmentation consistently outper-

Table 2: Results of AST-T5 on downstream tasks compared with reported results of established language models. Evaluation metrics align with those in Table 1. Our focus is primarily on models with similar sizes as AST-T5, specifically the "Base" models (110M to 230M parameters), while comparisons against larger models are depicted in Figure 3. Some models are either encoder-only or decoder-only and are thus not suited for certain tasks. These results are labeled with "N/A" in this table because they are not available in the literature.

| Model | Generation | | Transpilation | | Understanding | |
|---|---|---|---|---|---|---|
| | HumanEval | Concode | Bugs2Fix | Java-C# | Clone | Defect |
| CodeBERT | N/A | N/A | 16.4 / 5.2 | 59.0/58.8 | 96.5 | 62.1 |
| GraphCodeBERT | N/A | N/A | 17.3 / 9.1 | 59.4/58.8 | 97.1 | N/A |
| PLBART | N/A | 18.8 | 19.2 / 9.0 | 64.6/65.0 | 97.2 | 63.2 |
| CodeT5 | N/A | 22.3 | 21.6/14.0 | 65.9/66.9 | 97.2 | 65.8 |
| CodeT5+$_{\text{BASE}}$ | 12.0 | N/A | N/A | N/A | 95.2 | **66.1** |
| StructCoder | N/A | 22.4 | N/A | 66.9/68.7 | N/A | N/A |
| AST-T5 (Ours) | **12.8** | **22.9** | **23.8/16.1** | **68.9/72.3** | **98.6** | 65.8 |

forms the T5 baseline that uses Greedy Segmentation across all tasks. The advantage stems from the fact that AST-Aware Segmentation produces less fragmented and thus less noisy training inputs during pretraining. Given that most downstream tasks present coherent code structures, such as entire function definitions, the consistency upheld by AST-Aware pretraining aligns better with these structures, leading to improved generalization.

**AST-Aware Span Corruption further boosts generation performance.** A comparison between the second and third rows of Table 1 reveals an improvement when shifting from Vanilla T5 Span Corruption to our AST-Aware Subtree Corruption. This performance gain is especially notable in generation and transpilation tasks. Such enhancements stem from the ability of AST-Aware Subtree Corruption to guide the model in generating code with better coherence and structural integrity.

**Increasing masking ratio improves generation performance.** The typical span corruption mask ratio in T5 is set at 15%. Increasing this ratio could potentially enhance the model's generation capabilities, albeit potentially at the expense of understanding tasks. Essentially, a mask ratio of 100% would emulate a GPT-like, decoder-only Transformer. However, in our experiments (last two rows of Table 1), we observed that raising the mask ratio from 15% to 25% significantly improved generation capabilities without noticeably compromising performance in understanding tasks. Further analysis shows that increasing the masking ratio to 50% yields only a marginal improvement on HumanEval (from 12.8 to 13.0), while adversely impacting transpilation and understanding tasks. Thus, we settled on a 25% mask ratio for our AST-T5 model.

## 5.2 MAIN RESULTS

Table 2 shows AST-T5's performance on downstream tasks compared with previously published results of similarly sized models, specifically those within the "Base" scale (110M to 230M parameters). Figure 3 and Figure 4 extends this comparison, comparing AST-T5 with larger models using the HumanEval benchmark and the MBPP benchmark, respectively. These results show that:

**AST-T5 excels as a unified and parameter-efficient LM for various code-related tasks.** While comparable in size, AST-T5 consistently outperforms similar-sized models such as CodeT5 (Wang et al., 2021) and CodeT5+ (Wang et al., 2023) in code generation, transpilation, and understanding. Notably, while CodeT5 and CodeT5+ are models at the Base scale, they were evaluated across different tasks. Our model, AST-T5, outperforms the best results of these two models across multiple benchmarks at the same time. Moreover, Figure 3 highlights AST-T5's competitiveness against significantly larger models like GPT-J (Wang & Komatsuzaki, 2021) and LLaMa-7B (Touvron et al., 2023) on the HumanEval benchmark, underscoring our model's parameter efficiency.

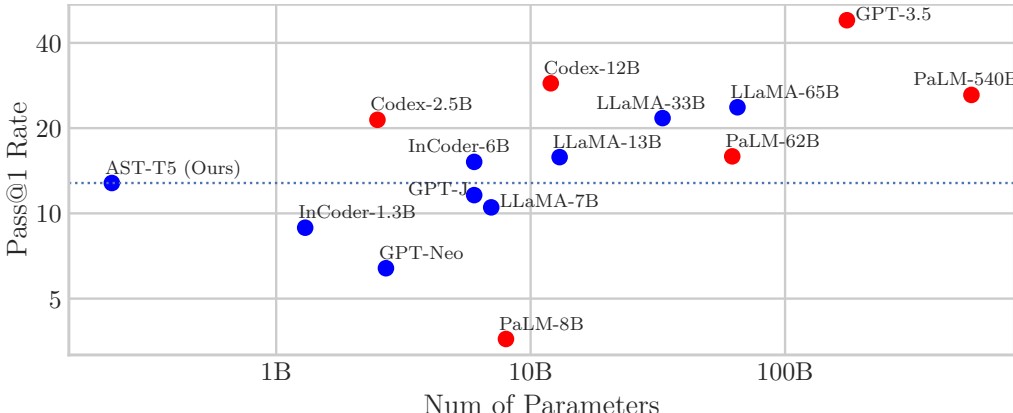

Figure 3: Visualization of AST-T5's performance on HumanEval compared to models exceeding 230M parameters and thus not detailed in Table 2. Each point on the scatter plot represents a model. The x-axis shows the parameter count in log-scale, while the y-axis shows the Pass@1 rate on HumanEval in log-scale. Model open-source status is color-coded: **blue** for open-source and **red** for proprietary.

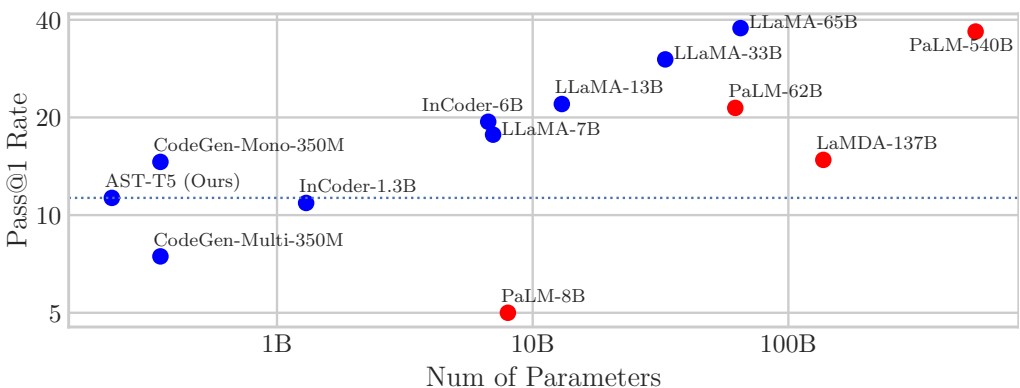

Figure 4: Visualization of AST-T5's performance on MBPP compared to other models. Each point on the scatter plot represents a model.

**AST-T5 exhibits unique strengths in transpilation through AST-awareness.**    Table 2 highlights AST-T5's superior performance in code-to-code transpilation tasks, showcasing gains a substantial gain of 2 to 5 points on Bugs2Fix and Java-C# transpilation. In transpilation, while surface-level code can exhibit significant variability, the intrinsic AST structures of the source and target often maintain a notable similarity. The capability of AST-T5 to exploit this structural similarity is crucial to its effectiveness. The benefits of being structure-aware are further exemplified by AST-T5's leading results in Clone Detection, where it surpasses CodeT5 by 3 points, because AST comparisons yield more precise insights than direct code comparisons.

## 6    Conclusion and Future Work

In this work, we present AST-T5, a novel pretraining paradigm that harnesses the power of Abstract Syntax Trees (ASTs) to boost the performance of code-centric language models. Using two structure-aware techniques, AST-T5 not only outperforms models of comparable size but also competes favorably against some larger counterparts. The simplicity of AST-T5 lies in its singular pretraining objective and its adaptability as a drop-in replacement for any encoder-decoder LM, highlighting its potential for real-world deployments. Moving forward, we aim to explore the scalability of AST-T5 by training larger models on more expansive datasets.

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

# A APPENDIX

## A.1 LIMITATIONS

AST-T5 is specifically designed to enhance code generation performance by exclusively masking code within AST subtrees during pretraining. While this specialized approach is advantageous for code generation tasks, it may result in suboptimal performance in natural language generation. Acknowledging this limitation, future versions of AST-T5 could investigate strategies such as masking docstrings and comments to broaden its applicability. This would potentially improve performance across various tasks, including code summarization.

## A.2 MORE ABOUT AST-AWARE SEGMENTATION

In Section 3.2, we use a dynamic programming algorithm to calculate the segmentation that results in the least number of AST structure breaks. A naive implementation of the DP algorithm is shown in Algorithm 3.

---

**Algorithm 3** Dynamic Programming in AST-Aware Segmentation (Before Optimization)

```
1    for k in range(1, m + 1):
2        for i in range(1, n + 1):
3            best_j = i - max_len
4            for j in range(i - max_len + 1, i):
5                if dp[k - 1, j] < dp[k - 1, best_j]:
6                    best_j = j
7            prev[k, i] = best_j
8            dp[k, i] = cost[i] + min_value
```

---

Denote the length of the code file (in tokens) by $n$. In the algorithm, $m$ denotes the maximum number of chunks that the file can be split into, which is approximately $n/\mathrm{max\_len}$. So this implementation has time complexity $O(mn \cdot \mathrm{max\_len}) = O(n^2)$, which is not feasible for longer code files. To optimize this algorithm, we use a monotonic queue to compute the sliding-window minimum, as described in Algorithm 1.

Each element is only pushed into and popped out of the monotonic queue once, so the time complexity of the optimized algorithm is $O(nm) = O(n^2/\mathrm{max\_len})$, making the algorithm $\tilde{1}000$x faster when $\mathrm{max\_len} = 1024$. This allows the algorithm to segment each code file with 100k tokens in milliseconds.

## A.3 PRETRAINING CORPORA

Our pretraining corpora consists of two parts: code and natural language, coming from three sources:

- **GitHub (408 GB):** The "GitHub repositories" public dataset available on Google Big-Query [3]. For pretraining, we use all code files in Python (70 GB), C/C++ (195 GB), Java (105 GB), C# (38 GB) from each repo with an open-source license that explicitly permits redistribution.

- **Wikipedia (16 GB):** A natural language corpus widely used for natural language pretraining.

- **OpenWebText (38 GB):** A natural language corpus used by Liu et al. (2019) to train language models.

## A.4 PRETRAINING HYPERPARAMETERS

Table 3 shows the pretraining hyperparameters for our proposed AST-T5 model.

---

[3]https://console.cloud.google.com/marketplace/details/github/github-repos

| | |
|---|---|
| Encoder Layers | 12 |
| Decoder Layers | 12 |
| Hidden Dimension | 768 |
| Peak Learning Rate | 2e-4 |
| Batch Size | 1,024 |
| Warm-Up Steps | 10,000 |
| Total Steps | 500,000 |
| Sequence Length | 1,024 |
| Mask Ratio | 25% |
| Min Subtree Corruption Threshold $\theta$ | 5 |
| Max Subtree Corruption Threshold $\theta$ | 100 |
| Relative Position Encoding Buckets | 32 |
| Relative Position Encoding Max Distance | 128 |
| Adam $\epsilon$ | 1e-6 |
| Adam $(\beta_1, \beta_2)$ | (0.9, 0.98) |
| Clip Norm | 2.0 |
| Dropout | 0.1 |
| Weight Decay | 0.01 |

Table 3: Pretraining hyperparameters for our AST-T5 model.

Table 4: Overview of our evaluation benchmarks detailing test set size, task type, and evaluation metric for each task. For MBPP, we follow Nijkamp et al. (2023) and evaluate our model on the entire "sanitized" subset without few-shot prompts. For evaluation metrics, "Pass@1" indicates code execution on unit-tests provided in the benchmark using a single generated code per example, with reported pass rates. "Exact Match" evaluates textual equivalence without execution by comparing two canonicalized code pieces. We omit "BLEU scores" because high BLEU values ($> 50$) can still correspond to unexecutable or significantly flawed code (Lu et al., 2021), which is not useful in real-world applications. We also discuss evaluation results using the CodeBLEU (Ren et al., 2020) metric in Appendix A.7.

| | Size | Type | Metric |
|---|---|---|---|
| HumanEval | 164 | Text-to-Code Generation | Pass@1 |
| MBPP | 427 | Text-to-Code Generation | Pass@1 |
| Concode | 2,000 | Text-to-Code Generation | Exact Match |
| Bugs2Fix | 12,379 | Code-to-Code Transpilation | Exact Match |
| Java-C# | 1,000 | Code-to-Code Transpilation | Exact Match |
| BigCloneBench | 415,416 | Code Understanding | F1 |
| Defect Detection | 27,318 | Code Understanding | Accuracy |

## A.5 EVALUATION BENCHMARKS

We evaluate AST-T5 across three types of tasks: text-to-code generation, code-to-code transpilation, and code understanding (classification). Our evaluation encompasses tasks from the CodeXGLUE meta-benchmark (Lu et al., 2021) and also includes HumanEval (Chen et al., 2021a) and MBPP (Austin et al., 2021). Specifically, for text-to-code generation, we assess performance using HumanEval, MBPP, and Concode (Iyer et al., 2018); for transpilation, we use CodeXGLUE Java-C# and Bugs2Fix (Tufano et al., 2019) for evaluation; and for understanding, we use Big-CloneBench (Svajlenko et al., 2014) and the Defect Detection task proposed by Zhou et al. (2019). Detailed metrics and statistics of these datasets are provided in Table 4.

## A.6 EVALUATION RESULTS ON MULTI-LINGUAL CODE GENERATION

Table 5 presents a comparative analysis of our AST-T5 model on Python and Java subsets of the multi-lingual HumanEval and MBXP benchmarks (Athiwaratkun et al., 2023). This analysis includes models such as BLOOM (BigScience, 2021), OPT (Zhang et al., 2022), and various con-

Table 5: Results of AST-T5 on multi-lingual HumanEval and MBXP compared with reported results of established language models. The evaluation metric is Pass@1.

| | #Params | HumanEval | | MBXP | |
|---|---|---|---|---|---|
| | | Python | Java | Python | Java |
| CodeGen-multi | 350M | 7.3 | 5.0 | 7.5 | 8.2 |
| CodeGen-mono | 350M | 10.3 | 3.1 | **14.6** | 1.9 |
| AST-T5 (Ours) | 226M | **12.8** | **10.6** | 11.3 | **9.8** |
| BLOOM | 7.1B | 7.9 | 8.1 | 7.0 | 7.8 |
| OPT | 13B | 0.6 | 0.6 | 1.4 | 1.4 |
| CodeGen-multi | 2B | 11.0 | 11.2 | 18.8 | 19.5 |
| CodeGen-mono | 2B | 20.7 | 5.0 | 31.7 | 16.7 |
| CodeGen-multi | 6B | 15.2 | 10.6 | 22.5 | 21.7 |
| CodeGen-mono | 6B | 19.5 | 8.7 | 37.2 | 19.8 |
| CodeGen-multi | 16B | 17.1 | 16.2 | 24.2 | 28.0 |
| CodeGen-mono | 16B | 22.6 | 22.4 | 40.6 | 26.8 |

Table 6: Results of AST-T5 on CONCODE with reported results of established language models. The evaluation metric is exact match score and CodeBLEU.

| | EM | CodeBLEU |
|---|---|---|
| GPT-2 | 17.4 | 29.7 |
| CodeGPT-2 | 18.3 | 32.7 |
| CodeGPT-adapted | 20.1 | 36.0 |
| PLBART | 18.8 | 38.5 |
| CodeT5-Small | 21.6 | 41.4 |
| CodeT5-Base | 22.3 | 43.2 |
| AST-T5 (Ours) | **22.9** | **45.0** |

figurations of CodeGen (Nijkamp et al., 2023), as reported in Athiwaratkun et al. (2023). Our results show AST-T5's superior performance across all benchmarks compared to the CodeGen-multi-350M. Notably, although CodeGen-mono-350M, tailored for Python, surpasses AST-T5 in the MBPP benchmark, it significantly underperforms in the Java subset. Furthermore, AST-T5, having 226M parameters, outperforms larger counterparts like BLOOM-7.1B and OPT-13B.

## A.7 EVALUATION RESULTS IN CODEBLEU

Table 6 presents the performance of various models on the Concode dataset using the CodeBLEU metric, as reported in (Wang et al., 2021). CodeBLEU, specifically designed for evaluating code synthesis, computes a weighted average of three scores: textual match (BLEU), AST match, and Data Flow Graph (DFG) match. Our findings show a clear correlation between CodeBLEU and exact match scores.

