# OpenReview forum: "AST-T5: Structure-Aware Pretraining for Code Generation and Understanding"
_ICLR.cc/2024/Conference — Submitted to ICLR 2024_

### Official Review · Reviewer_vdfV · 2023-10-31

**Soundness:** 3 good
**Presentation:** 3 good
**Contribution:** 3 good
**Rating:** 5
**Confidence:** 4

**Summary:**

- This work proposes AST-aware segmentation of code into chunks to smartly chunk code into contexts causing the least disruption of syntax trees.
- Building on top of AST-aware segmentation, this work proposes an AST-aware subtree corruption pre-training objective instead of the standard random corruption objective to mask well-formed code segments.
- Using this sole pre-training objective, this work pre-trains a T5 model of 226M parameters on code data from GitHub and NL data from Wikipedia and OpenWebText.
- Evaluating AST-T5 on downstream tasks such as code transpilation, code generation, and code understanding, the work shows better performance of the AST-T5 model as compared to other comparable models.
- Through ablation studies, the work demonstrates the benefits of AST segmentation and AST subtree corruption separately.

**Strengths:**

The proposed method is a simple yet useful way of utilizing code syntax trees in model pre-training that results in better performance. I do believe that utilizing AST to better design pre-training objectives code has a lot of merit, and this paper presents an interesting approach to how to think about code segmentation and masking, and how it affects model performance.

**Weaknesses:**

- The work refers to trees created by the tree-sitter library as ASTs, but the tree-sitter library actually creates a Concrete Syntax Tree and not AST. While the two are related, CSTs are precursors to ASTs and are more verbose. See [1], [2], [3] for more details.
- What is the rationale behind including explicit NL data in the pre-training corpus? Prior works in this space such as CodeT5, PLBART, and GraphCodeBERT all have only code data in their pre-training mix. Is this a side-effect of the AST-aware masking strategy, where comments in code are not masked, and thus rather than including a new task - such as the Bimodal dual generation task in CodeT5 - this work requires NL data?
- I have the impression that the AST Subtree Corruption objective is more important than AST-aware segmentation because, for the defined objective, the model is learning to unmask spans masked in an AST-aware manner through the AST Subtree Corruption. The remaining context (code), even if incomplete from an AST point-of-view might not be affecting the learning significantly. Is it possible to do an ablation study with base T5 + AST Subtree Corruption to assess the performance? Since this work utilizes tree-sitter, which is an incremental parsing library, it is possible to parse even incomplete code.
- For Concode, Bugs2Fix, and Java-C# translation tasks, the work reports the Exact Match (EM) score alone. EM is not an ideal measure for code as it measures string equivalence. Other fuzzy evaluation metrics such as CodeBLEU [4] or CodeBERTScore [5] scores might be a better fit for these tasks.
- While the work does a good job of covering a decent number of downstream tasks, any specific reason why Code Summarization tasks were not included? The pre-training dataset for the model does include explicit NL data, so I expected to see code-to-text tasks included as well.
- In section 4, the authors mention utilizing PromptSource to create prompt templates for fine-tuning. I might be wrong, but it is my understanding that prompts are more useful in zero/few-shot settings. If the authors are fine-tuning the model, why is a prompt required? Why can't the model be fine-tuned on input-output pairs directly?



**References:**

[1] - https://en.wikipedia.org/wiki/Abstract_syntax_tree

[2] - https://en.wikipedia.org/wiki/Parse_tree

[3] - https://eli.thegreenplace.net/2009/02/16/abstract-vs-concrete-syntax-trees/

[4] - https://arxiv.org/abs/2009.10297

[5] - https://github.com/neulab/code-bert-score

**Questions:**

mentioned above.

---

> ### Author Response · Authors · 2023-11-22
> **Response to Reviewer vdfV (1/2)**
>
> **W1. Misused terms: tree-sitter library actually creates a Concrete Syntax Tree and not AST**
>
> Thank you for highlighting the distinction between Concrete Syntax Trees (CSTs) and Abstract Syntax Trees (ASTs). We acknowledge that Tree-Sitter produces CSTs, which, while related, are indeed more verbose and concrete compared to ASTs. Our model, AST-T5, indeed relies on CST parsing rather than AST-level abstractions.
>
> To address this, we will amend our manuscript to replace all instances of "AST" with "Parse Tree" and "AST-aware" with "PT-aware" in the next version. We apologize for not making these changes in the current draft, as the title cannot be modified during the rebuttal phase, and we aim to maintain consistency in our discussion. Thank you for your valuable suggestions in improving the accuracy and clarity of our paper!
>
> **W2. What is the rationale behind including explicit NL data in the pre-training corpus?**
>
> We thank the reviewer for their insightful attention to the inclusion of explicit natural language (NL) data in our pre-training corpus. It's important to note that NL data has been a key component in training code-related models, as evidenced by various models in the field. For example, Codex leverages a rich NL foundation by fine-tuning from GPT-3, and models like CodeT5 and GraphCodeBERT use NL-PL bimodal corpora, including CodeSearchNet, to enhance their training datasets. Moreover, the practice of using the entire GitHub dump, which encompass NL data from sources such as Markdown, JavaDoc, Plain Text, or HTML, is a standard approach in this domain (refer to https://huggingface.co/datasets/bigcode/the-stack-dedup/tree/main/data/markdown). Additionally, CodeLLaMa [2] explicitly incorporate web-based NL data, such as StackOverflow discussions, into their training processes. Therefore, the use of NL data is a common thread across existing models.
>
> Our inclusion of explicit NL data from external datasets compensates for the limitations in our data crawling pipeline, which focuses on programming languages such as Python, Java, C, C++, and C# through file extensions like ".py" or ".cpp". This approach, while effective for sourcing code, often misses rich NL content in Markdown and plain text files, essential in training processes and used by comparable models in the field. By integrating additional NL datasets, we ensure our pretraining corpus is comprehensive, addressing this critical gap. This decision is independent of our AST-aware masking strategy.
>
> **W3. Is it possible to do an ablation study with base T5 + AST Subtree Corruption to assess the performance?**
>
> Thank you for your suggestion regarding the additional experiment for "Base T5 + AST-Aware Subtree Corruption". However, AST-Aware Subtree Corruption, as outlined in Algorithm 2, is dependent on AST-Aware Segmentation. It operates recursively on each subtree within a segmented chunk. Without the AST-Aware Segmentation, the model would encounter highly fragmented spans, because standard Greedy Segmentation could result in a code segment comprising numerous disjoint subtrees. This fragmentation would likely degrade the performance of such "Base T5 + AST-Aware Subtree Corruption" model.
>
> **W4. Other fuzzy evaluation metrics such as CodeBLEU [4] or CodeBERTScore [5] scores might be a better fit for some tasks such as Concode.**
>
> Thank you for your valuable suggestion to incorporate alternative evaluation metrics. In response, we have included CodeBLEU scores in Appendix A.7 for Concode, because this metric has also been used in prior works like CodeT5. The result is also shown here, the reported numbers of other models are from [1]:
>
> |                 | **Exact Match** | **CodeBLEU** |
> |-----------------|----------------:|-------------:|
> | GPT-2           |            17.4 |         29.7 |
> | CodeGPT-2       |            18.3 |         32.7 |
> | CodeGPT-adapted |            20.1 |         36.0 |
> | PLBART          |            18.8 |         38.5 |
> | CodeT5-Small    |            21.6 |         41.4 |
> | CodeT5-Base     |            22.3 |         43.2 |
> | AST-T5 (Ours)   |        **22.9** |     **45.0** |
>
> Our analysis reveals a strong correlation between CodeBLEU and Exact Match (EM) scores, and AST-T5 outperforms baselines in both metrics, underscoring the robustness of our model's performance. Regarding CodeBERTScore, we noted its dependency on a specific vocabulary, limiting its applicability compared to the baselines we have used, such as CodeT5.
>
> [1] CodeT5: Identifier-aware Unified Pre-trained Encoder-Decoder Models for Code Understanding and Generation, https://arxiv.org/abs/2109.00859
>
> [2] Code Llama: Open Foundation Models for Code, https://arxiv.org/abs/2308.12950

---

> ### Author Response · Authors · 2023-11-22
> **Response to Reviewer vdfV (2/2)**
>
> **W5. Any specific reason why Code Summarization tasks were not included?**
>
> Thank you for your insights! Yes, this is indeed a limitation of our work. AST-T5, as it currently stands, excels in code generation by using a unique masking strategy within AST subtrees during pretraining. This specificity, while beneficial for code-centric tasks, may not yield the best results in natural language generation. Acknowledging this limitation, future versions of AST-T5 could investigate strategies such as masking docstrings and comments to broaden its applicability. This would potentially improve performance across various tasks, including code summarization. We have also included the discussion of limitations of AST-T5 in the revised manuscript, specifically in Appendix A.1.
>
> **W6. If the authors are fine-tuning the model, why is a prompt required? Why can't the model be fine-tuned on input-output pairs directly?**
>
> Thank you for your insightful question regarding the use of prompts in our fine-tuning process. Our approach, diverging from CodeT5's single-task fine-tuning, uses a multi-task framework similar to those in renowned NLP models like T5 and T0. This necessitates prompts to differentiate between various tasks, ensuring consistency across all models, including AST-T5 and baselines, for fair experimental comparison.
>
> Moreover, prompts in our fine-tuning cater to challenges in datasets like HumanEval, which lack training data. By applying prompt-based fine-tuning on CodeSearchNet, AST-T5's performance on HumanEval is notably enhanced. While CodeT5+ also adapts to NL-to-Code tasks using CodeSearchNet, it differs by using an alternate loss function, not prompts. However, prompt usage is a secondary aspect of our evaluation protocol; the primary focus and contribution of our paper is the AST-Awareness, as clearly demonstrated by the results in Table 1.

---

### Official Review · Reviewer_s5j8 · 2023-10-31

**Soundness:** 3 good
**Presentation:** 3 good
**Contribution:** 3 good
**Rating:** 6
**Confidence:** 5

**Summary:**

The authors introduce a new way of pre-training generative models for code which has two innovations:

First, when dividing long source code files into smaller segments for training purposes, the authors choose segmentation boundaries that align, as much as possible, with the abstract syntax tree (AST) for the code.

Second, when masking out spans of text, where the model is trained to reconstruct the masked spans, they mask out spans that correspond to nodes in the AST.

Experiments show that these two changes to the training task result in significant improvements in model quality.  No changes are made to the transformer model architecture, or to the training data set.

The authors compare the performance of a single model which is pretrained with AST-aware segmentation and masking to the performance of the same sized model which is pretrained with conventional segmentation and masking.  They also compare their model against other models (with different sizes and architectures)  in the literature.

**Strengths:**

The paper is well written and the ideas are presented clearly.  The experiments also seem to be well-designed, and adequately support the authors conclusions.

The most interesting thing about this paper is that the authors are able to achieve significant improvements, just by changing segmentation and masking.  There are no changes to the actual training data or to the transformer architecture.  The use of code structure has, IMO, been underutilized in code LLMs to date, so this paper outlines a simple and effective technique that other practitioners can implement.

**Weaknesses:**

The main weakness of the paper is that it only investigates these two simple changes.  There are many other previously published ways of utilizing AST structures in a transformer architecture, e.g. tree-position embeddings, augmenting the attention matrix with tree/graph structure, etc.  Many of these changes would not be hard to implement.  I would have been very interested to see a more comprehensive comparison of various techniques.  E.g. how does masking (in this paper) compare against providing tree-position embeddings for the AST?  Does AST-aware attention eliminate the benefit of AST-aware masking, or do they compliment each other?

Experimentally, this paper uses a context length of 1024 tokens, which is very small for code.  The benefit of AST-aware segmentation will diminish with increasing context length, so some of these results may not apply to larger models.

The related work section also needs to be fleshed out, in particular with regards to "leveraging code structure in pretraining."

**Questions:**

There is less need for good segmentation boundaries when training with longer contexts.  Have you attempted to see whether your results still hold for model that are trained on longer contexts, e.g. 8k or 32k tokens?

---

> ### Author Response · Authors · 2023-11-22
> **Response to Reviewer s5j8 (1/2)**
>
> **W1. About other previously published ways of utilizing AST structures**
>
> We acknowledge the potential benefits of various AST-aware techniques in Transformer architecture. Indeed, methods like tree-position embeddings and AST-aware attention mechanisms have shown promise. However, these approaches often come with architectural changes and assumptions that may not align with all use cases, particularly those involving incomplete or erroneous code, which is common in real-world applications like bug fixing or code completion.
>
> Our decision to focus on AST-T5's simple yet effective approach was strategic. By avoiding complex architectural changes and maintaining compatibility with existing T5 variants, AST-T5 is designed for seamless integration and broader applicability. This simplicity also negates the need for parsing or static analysis in downstream tasks, which is a crucial factor for practicality in varied coding scenarios.
>
> While integrating additional AST-aware techniques like tree-position embeddings could potentially enhance the model, such modifications would compromise its general applicability and ease of use. AST-T5's focus on pretraining with AST-awareness, rather than relying on complex runtime processes, represents a balance between capturing code's structural nuances and maintaining a flexible, adaptable framework suitable for a wide range of applications.
>
> We have expanded our discussion on related work in the manuscript's revised "Related Work" section.
>
> **W2. This paper uses a context length of 1024 tokens. The benefit of AST-aware segmentation will diminish with increasing context length**
>
> We appreciate your attention to the context length in AST-T5. Notably, AST-T5 uses a context length of 1024 tokens for both source and target. This exceeds the context lengths of our main baselines, CodeT5 and CodeT5+, which are 512/256 and 768/600 tokens respectively.
>
> The AST-Aware segmentation at the core of AST-T5 is pivotal in maintaining the structural integrity of code during pretraining, offering two benefits:
>
> 1. *Enhanced Pretraining Efficiency*: The structured nature of inputs, thanks to AST-aware segmentation, boosts pretraining efficiency. While this advantage may diminish with longer context lengths (e.g., 2k, 8k, 32k tokens), its impact is significant in our current setup.
>
> 2. *Reduced Mismatch with Downstream Tasks*: AST-aware segmentation ensures a closer match between pretraining inputs and the complete code structures encountered in downstream tasks, like function definitions. This alignment minimizes the pretraining-evaluation mismatch, a benefit that remains crucial regardless of context length.
>
> Although the first benefit might reduce as context lengths increase, the second benefit remains highly relevant. We acknowledge the value of experimenting with longer context lengths, and we plan to explore this in future work, once we have sufficient computational resources.

---

> ### Author Response · Authors · 2023-11-22
> **Response to Reviewer s5j8 (2/2)**
>
> **W3. The related work section also needs to be fleshed out, in particular w.r.t. "leveraging code structure in pretraining."**
>
> Thank you for highlighting the need for a more comprehensive discussion of leveraging code structure in pretraining in the related work section. In response to your insightful suggestion, we have extensively expanded this paragraph, doubling its length to provide more discussions and comparisons about relevant studies. Here is the updated version of this part:
>
> >    **Leveraging Code Structure in Pretraining.** Code differs from natural language in two key aspects: its executability and strict structural syntax. Previous research leveraged execution traces for improving model performance (Chen et al., 2018; 2021b; Shojaee et al., 2023) but this approach faces scalability challenges when applied to large, web-crawled code datasets used in pretraining. Regarding code’s structured nature, various studies have integrated syntactic elements into neural network models. Li et al. (2018), Kim et al. (2021) and Zugner et al. (2021) add AST-Aware attention mechanisms in their models, while Alon et al. (2020) and Rabinovich et al. (2017) focus on modeling AST node expansion operations rather than traditional code tokens. In parallel, Guo et al. (2021) and Allamanis et al. (2017) explore DFG-Aware attention mechanisms and Graph Neural Networks (GNNs), to interpret code based on its Data Flow Graph (DFG). StructCoder (Tipirneni et al., 2023) enriches the code input by appending AST and DFG as additional features. These methods, however, necessitate parsing or static analysis for downstream tasks, which is less feasible for incomplete or incorrect code scenarios like bug fixing.
> >
> >
> >
> >    Our work, AST-T5, aligns with methods that utilize code structure only in pretraining, like DOBF (Roziere et al., 2021) and CodeT5 (Wang et al., 2021), which obfuscate inputs to force the model to grasp abstract structures. Our approach uniquely diverges by using AST-driven segmentation and masking in T5 span corruption during pretraining. This novel approach offers a more refined pretraining signal compared to structure-agnostic T5, equipping our model to proficiently encode and generate semantically coherent code structures.
>
> We believe these enhancements not only address your concerns but also enrich the clarity of the manuscript. We are grateful for your feedback and hope that our revisions meet your expectations.

---

### Official Review · Reviewer_dEhq · 2023-11-09

**Soundness:** 3 good
**Presentation:** 3 good
**Contribution:** 3 good
**Rating:** 6
**Confidence:** 4

**Summary:**

This paper presents a pretraining paradigm for code models consisting using AST-aware methods. Specifically, it presents two main techniques: (1) A method of segmenting examples to minimize AST breakage and (2) a span corruption technique that minimizes AST subtrees cutting instead of randomly choosing.

Update: My score has been adjusted to reflect the concerns addressed by the authors.

**Strengths:**

While AST information use in code LLMs is not novel, the combination of using this information in segmentation and T5-style span corruption in pretraining is original and interesting.

**Weaknesses:**

- Performance comparisons to other models require more supporting evidence
  - Given the lack of models for comparison at this scale (in particular for generation tasks with more N/A than results in the chart), showing these results at a scale with more models to compare directly against would have made the performance differences more abundantly clear and understanding the scaling characteristics would make for a much more compelling argument.
  - HumanEval has been known to have been in public datasets (such as Github Archive) commonly used for training code LLMs (https://arxiv.org/abs/2306.11644) and being that it is an incredibly small dataset of questions and only Python, it would a stronger argument for performance to show the Pass@1 vs Model Size graph for other common reported downstream evaluations to support the claim of "competitiveness against larger models". I understand that there are not many code generation evaluations that are common between most models, but I would suggest at least adding MBPP (present in Llama, PaLM, GPTs, etc) and consider showing performance across more languages through benchmarks such as MBXP and Multilingual HumanEval (https://arxiv.org/abs/2210.14868).
- This sentence in the abstract: `Structure-awareness makes it particularly powerful in code-to-code tasks, surpassing CodeT5 by 2 points in Bug Fixing and 3 points in Java-C# Transpilation.` needs to be contextualized better or a different summary of results should be presented. The significance of these results are ambigious.
- Typo: "avilable" typo in page 9 last line of first paragraph

**Questions:**

1. What are some potential limitations of using this method?

2. Were masking ratios tested between 25% and 100%? If 25% showed no degradation in understanding tasks, why not go higher?
From paper:
```
we observed that raising the mask ratio from 15% to 25% significantly improved generation capabilities without noticeably compromising performance in understanding tasks. Thus, we settled on a 25% mask ratio for our AST-T5 model.
```
3. Code completion (generation with both prefix and suffix context) has become a very common capability in code models and the span corruption objective is effectively code completion. Was there work done on evaluating this model for code completion tasks? Why or why not?

4. The abstract mentioned anonymized model + code, where is this available?

---

> ### Author Response · Authors · 2023-11-22
> **Response to Reviewer dEhq (1/3)**
>
> **W1.1. Lack of models for comparison at this scale**
>
> Regarding the scale of model comparison, the limited computational resources indeed restricted our capacity for training larger models. Despite this, our study ensures a fair and rigorous comparison as detailed in Table 1, where all models, including AST-T5, were trained using the same dataset and environment. This approach enables a meaningful evaluation of AST-T5's performance against similar-sized models, showing its effectiveness in code-related tasks. Our focus on methodological rigor over scale underscores the value of AST-T5’s contributions.
>
> **W1.2. Potential data contamination of HumanEval and the need for results on more benchmarks**
>
> Thank you for your insightful comments on dataset overlap and benchmarks. Our pretraining data, sourced from Google BigQuery's GitHub dump as of Summer 2021, predates the release of Codex and HumanEval, thus mitigating concerns about dataset overlap.
>
> Incorporating your valuable suggestions, we've added the results on the MBPP benchmark in Figure 4, and results on the Java split of the Multi-lingual HumanEval and MBXP benchmarks in Appendix A.6. Currently, AST-T5 supports Python, Java, C/C++, and C#, and we have already shown good results while running on these languages. Future work will extend our pretraining dataset to include all languages in these benchmarks. Due to the absence of results from models at a comparable scale (226M parameters), we compare AST-T5 with both the 350M-parameter CodeGen-multi (multi-lingual) and CodeGen-mono (Python-specialized) models, as well as larger models for a broader context.
>
>
> Our results are summarized below:
>
> Result on MBPP:
>
> |               | **#Params** | **MBPP** |
> |---------------|-------------|----------|
> | CodeGen-multi | 350M        | 7.5      |
> | AST-T5 (Ours) | 226M        | **11.3** |
> |               |             |          |
> | PaLM          | 8B          | 5.0      |
> | PaLM          | 62B         | 21.4     |
> | PaLM          | 540B        | 36.8     |
> | CodeGen-mono  | 350M        | 14.6     |
> | InCoder       | 1.3B        | 10.9     |
> | InCoder       | 6.7B        | 19.4     |
> | LaMDA         | 137B        | 14.8     |
> | LLaMA         | 7B          | 17.7     |
> | LLaMA         | 13B         | 22.0     |
> | LLaMA         | 33B         | 30.2     |
> | LLaMA         | 65B         | 37.7     |
>
> Result on multi-lingual HumanEval and MBXP, compared with reported numbers in [1]:
>
> |               | **#Params** | **HumanEval** | **HumanEval** | **MBXP** | **MBXP** |
> |---------------|-------------|---------------|---------------|----------|----------|
> |               |             | Python        | Java          | Python   | Java     |
> | CodeGen-multi | 350M        | 7.3           | 5.0           | 7.5      | 8.2      |
> | CodeGen-mono  | 350M        | 10.3          | 3.1           | **14.6**     | 1.9      |
> | AST-T5 (Ours) | 226M        | **12.8**      | **10.6**      | 11.3     | **9.8**  |
> |   |  |  | |  |   |
> | BLOOM         | 7.1B        | 7.9           | 8.1           | 7.0      | 7.8      |
> | OPT           | 13B         | 0.6           | 0.6           | 1.4      | 1.4      |
> | CodeGen-multi | 2B          | 11.0          | 11.2          | 18.8     | 19.5     |
> | CodeGen-mono  | 2B          | 20.7          | 5.0           | 31.7     | 16.7     |
> | CodeGen-multi | 6B          | 15.2          | 10.6          | 22.5     | 21.7     |
> | CodeGen-mono  | 6B          | 19.5          | 8.7           | 37.2     | 19.8     |
> | CodeGen-multi | 16B         | 17.1          | 16.2          | 24.2     | 28.0     |
> | CodeGen-mono  | 16B         | 22.6          | 22.4          | 40.6     | 26.8     |
>
> Despite its smaller size of 226M parameters, AST-T5 outperforms larger models like CodeGen-multi-350M, PaLM-8B, and InCoder-1.3B on MBPP, showing its efficiency. While CodeGen-mono-350M leads in MBPP because it is CodeGen-multi-350M further finetuned on Python, AST-T5, as a multi-lingual model, outperforms models like CodeGen-mono/multi-350M, BLOOM, and OPT in the Multilingual HumanEval and MBXP-Java benchmarks, underscoring its effectiveness in multiple programming languages.
>
> [1] Multi-lingual Evaluation of Code Generation Models, http://arxiv.org/abs/2210.14868
>
> **W2. Writing of the abstract**
>
> Thank you for pointing out the need for clearer contextualization in our abstract! To address this, we have revised the sentence to: “Structure-awareness makes AST-T5 particularly powerful in code-to-code tasks, surpassing CodeT5 by 2 points in exact match score for the Bugs2Fix task and by 3 points in exact match score for Java-C# Transpilation in CodeXGLUE.” This revision aims to provide a more context of our results, clearly indicating the metrics used and the specific tasks where improvements were observed. Thank you for helping us improve the clarity of our abstract!
>
> **W3. Typo**
>
> We apologize for the typo and we have fixed it in the updated manuscript.

---

> ### Author Response · Authors · 2023-11-22
> **Response to Reviewer dEhq (2/3)**
>
> **Q1. What are some potential limitations of this method?**
>
> We appreciate the opportunity to discuss the limitations of our work. AST-T5, as it currently stands, excels in code generation by using a unique masking strategy within AST subtrees during pretraining. This specificity, while beneficial for code-centric tasks, may not yield the best results in natural language generation. Acknowledging this limitation, future versions of AST-T5 could investigate strategies such as masking docstrings and comments to broaden its applicability. This would potentially improve performance across various tasks, including code summarization. We have also included the discussion of limitations of AST-T5 in the revised manuscript, specifically in Appendix A.1.
>
> **Q2. Were masking ratios tested between 25% and 100%? Why not go higher than 25%?**
>
> Thank you for your insightful question regarding the masking ratio in AST-T5. Masking ratio is an important hyperparameter in T5-style pretraining. Indeed, a 100% mask ratio would essentially make the model equivalent to a GPT-like, decoder-only architecture, underutilizing the encoder's parameters. Conversely, a 0% mask ratio is infeasible as it would negate any loss function application, rendering the model untrainable.
>
> We rigorously tested various masking ratios, including 15%, 25%, and, as newly added per your suggestion, 50%. The results of the 50% mask ratio experiment are now added to Table 1 of our paper and also included below. We observed that increasing the masking ratio from 25% to 50% led to a marginal improvement in HumanEval performance (improving Pass@1 from 12.8 to 13.0). However, this increase adversely affected both transpilation and understanding tasks.
>
> |                           | **HumanEval** | **Concode** |  **Bugs2Fix** |       **Java-C#** | **Clone** | **Defect** |  **Avg** |
> |---------------------------|--------------:|------------:|--------------:|------------------:|----------:|-----------:|---------:|
> | T5                        |           5.2 |        18.3 |     21.2/13.8 |         65.5/68.4 |      96.9 |       64.1 |     44.2 |
> | + AST. Segmentation       |           7.2 |        20.2 |     22.5/15.1 |         66.3/69.3 |      98.3 |       65.9 |     45.7 |
> | + AST. Subtree Corrupt    |           9.6 |        22.1 | 23.3/**16.5** |         67.3/72.2 |  **98.6** |   **66.0** |     47.0 |
> | + Mask 25\% (AST-T5)      |          12.8 |    **22.9** | **23.8**/16.1 | **68.9**/**72.3** |  **98.6** |       65.8 | **47.7** |
> | + Mask 50\% (Newly Added) |      **13.0** |        22.0 |     21.9/15.0 |         66.5/70.1 |      97.1 |       64.2 |     46.2 |
>
> Our analysis suggests that the transpilation and understanding tasks particularly benefit from a strong bidirectional Transformer encoder. A higher mask ratio, while potentially strengthening the decoder, correspondingly weakens the encoder. Therefore, we concluded that a 25% mask ratio offers the most balanced performance across various code-related tasks, without compromising the model's overall efficacy in transpilation and understanding.
>
> **Q3. Was there work done on evaluating this model for code completion tasks? Why or why not?**
>
> We acknowledge the relevance of the evaluation of AST-T5 on code completion tasks. However, the existing benchmarks for code completion, notably CodeXGLUE and HumanEval Fill-In-the-Middle (FIM) [1], present inherent biases and practical limitations when applied to our model. Specifically, CodeXGLUE's evaluation depends heavily on its predefined token vocabulary, disadvantageous to models like ours that uses a different Byte Pair Encoding (BPE) vocabulary. More critically, the FIM benchmark's approach of masking individual lines can disrupt the integrity of Abstract Syntax Tree (AST) structures. For instance, in the following code:
>
> ```python
> for x in l:
>     if x % 2 == 0:
>         print(x)
> ```
>
> masking a line such as `    if x % 2 == 0:` in a Python loop breaks the AST, as the line is only part of an AST node (the `if` node). Doing so puts AST-aware models at a disadvantage as the key idea behind such models is to treat code as structured units based on its node type (if, while, for, etc) instead of collections of tokens as in typical NLP setting. Masking individual lines is less representative of practical scenarios compared to masking AST nodes like `x % 2 == 0` or `x % 2`, where our model is likely to do better.
>
> To address these limitations, we adapted the single-line infilling task from [1] to only mask each line only when it contains a complete AST node (e.g., a full assignment statement). This modification yielded a pass@1 rate of 54.2% for AST-T5. In contrast, CodeT5+-220M achieved a pass@1 rate of 18.5% under the same conditions. The result shows the effectiveness of AST-T5 in a more relevant setting, which we will explore more later on.
>
>
> [1] Efficient Training of Language Models to Fill in the Middle, https://arxiv.org/abs/2207.14255

---

> ### Author Response · Authors · 2023-11-22
> **Response to Reviewer dEhq (3/3)**
>
> **Q4. The abstract mentioned anonymized model + code, where is this available?**
>
> The codebase has now been uploaded to the supplementary material section of our submission. We will make it public as we publish the paper. As for the model, we are currently in the process of merging our code into HuggingFace to facilitate its integration. We will release our model on the HuggingFace Model Hub. We appreciate your patience and interest in our work!

---

> > ### Comment · Reviewer_dEhq · 2023-11-23
> > **Thank you for the detailed responses**
> >
> > Thank you for the answering my questions, addressing my concerns, and including additional benchmarks. I've adjusted my score accordingly. I look forward to seeing the final open sourced model.

---

> > > ### Author Response · Authors · 2023-11-23
> > > **Thank you for your reassessment!**
> > >
> > > We sincerely appreciate your reassessment and the increase in score after our rebuttal. It is heartening to see that the additional benchmarks and responses to your questions were helpful. We're eager to finalize and open-source our model. Thank you again for your time and effort in reviewing our paper!

---

### Author Response · Authors · 2023-11-22
**General Response**

We sincerely thank all reviewers for their insightful and constructive feedback. We are grateful for the reviewers’ recognition of our work in various aspects:

1. The novelty of our AST-aware Segmentation and Span Corruption method (Reviewers dEhq and vdfV).
2. The simplicity and effectiveness of our model, AST-T5, and our proposed techniques, (Reviewers s5j8 and vdfV).
3. The solid design of our experiments, which adequately back our conclusions (Reviewer s5j8).
4. The clarity of our paper's presentation (Reviewer s5j8).

Following reviewers' suggestions, we have implemented the following revisions, with changes marked in blue in our updated manuscript:

1. **MBPP**: As suggested by Reviewer dEhq, we added the MBPP benchmark to Figure 4. AST-T5, despite its smaller size (226M parameters), demonstrates better performance (11.3% pass@1) over larger models like CodeGen-multi-350M (7.5% pass@1), PaLM-8B (5.0 pass@1), and InCoder-1.3B (10.9 pass@1) in this benchmark, showcasing its efficiency.

2. **Multi-Lingual Evaluations**: As suggested by Reviewer dEhq, we expanded our evaluations to include the Java split of the Multilingual HumanEval and MBXP benchmarks (detailed in Appendix A.6). AST-T5 outperforms models such as CodeGen-multi-350M, BLOOM, and OPT, thus highlighting its effectiveness in diverse programming languages.

3. **50% Mask Ratio**: As suggested by Reviewer dEhq, we added a new setup in Table 1, similar to AST-T5 but with the “mask ratio” hyperparameter set to 50%. Results show that increasing the mask ratio from 25% to 50% slightly improved HumanEval performance (from 12.8 to 13.0 pass@1) but negatively impacted transpilation and understanding tasks. Therefore, we set the mask ratio to 25% because it offers the most balanced performance across various code-related tasks.

4. **CodeBLEU Metric**: As suggested by Reviewer vdfV, we included an evaluation using the CodeBLEU metric for Concode in Appendix A.7, and made comparisons with the results reported in the CodeT5 paper. Our analysis reveals a strong correlation between CodeBLEU and Exact Match (EM) scores, and AST-T5 outperforms baselines in both metrics, underscoring the robustness of our model's performance.

5. **Codebase Release**: Our codebase is now part of the supplementary material, and we will make it public upon publication. We are also integrating our model into the HuggingFace platform for public access via the HuggingFace Model Hub.

6. **Expanded Related Work**: As suggested by Reviewer s5j8, we expanded the paragraph on "leveraging code structure in pretraining" in the "related work" section, which provides more discussions and comparisons about the approaches of each related work.

7. **Limitations**: As suggested by Reviewers dEhq and vdfV, we included a discussion on the limitations of our method in Appendix A.1—the AST-aware approaches are beneficial for code-centric tasks, yet may not yield the best results in natural language generation.

8. **Writing and Presentation**: As pointed out by Reviewer dEhq, we addressed the typos and clarified an ambiguous sentence in the abstract.

9. **Reorganization**: Due to page limitations, we moved the table of dataset statistics to the Appendix.

We extend our sincere thanks once again to all reviewers. Your constructive feedback has been invaluable in guiding us to refine and enhance the quality of our paper.

In the following sections, we address the specific concerns and questions raised by each reviewer.

---

### Meta-Review · Area_Chair_vh77 · 2023-11-30

**Metareview:**

This paper presents a technique for AST-driven pretraining example generation. The key idea is to use the code’s AST to segment code and to create valid syntactic spans to corrupt.

### Strengths:
- Simple method and easy method.
- Good empirical results for CodeT5-style architectures showing improved performance wrt to model parameter count.
- A wide range of target evaluation tasks.

### Weaknesses:
- Some relevant design space is left unexplored.
- While it is clear that the presented method drastically improves T5-style architectures, it is unclear if the improvements of this technique:
  - Transfer to decoder-only models (such as DOBF and InCoder-style architectures).
  - Whether the effects of improved performance due to the presented method diminish (or not) with model size.

While there is no reason to believe that the presented technique does _not_ generalize to other models/scales, there is also no experimental evidence for this. Thus, accepting this work risks straying the community towards a wrong direction.

Despite the above, it is appreciated that the authors may not have the computational resources to test the scaling behavior of the technique. In that case, I would still encourage them to test if their method is applicable to infilling or UL2 [a] style objectives of a “reasonably" small sized models.

[a] https://arxiv.org/pdf/2205.05131.pdf

**Justification For Why Not Higher Score:**

The generality of this technique is not tested, which could mislead the community.

**Justification For Why Not Lower Score:**

N/A

---

### Decision · Program_Chairs · 2024-01-16

Reject